# Changes in the Suicide Rate of Older Adults According to Gender, Age, and Region in South Korea from 2010 to 2017

**DOI:** 10.3390/healthcare10112333

**Published:** 2022-11-21

**Authors:** Kyu-Hyoung Jeong, Ji-Yeon Yoon, Seoyoon Lee, Sunghwan Cho, Hyun-Jae Woo, Sunghee Kim

**Affiliations:** 1Department of Social Welfare, Semyung University, Jecheon 27136, Republic of Korea; 2Institute for Life and Culture, Sogang University, Seoul 04100, Republic of Korea; 3Interdisciplinary Graduate Program in Social Welfare Policy, Yonsei University, Seoul 03722, Republic of Korea; 4School of Social Work, Virginia Commonwealth University, Richmond, VA 23223, USA; 5Health & Welfare Committee, Seoul Metropolitan Council, Seoul 04519, Republic of Korea

**Keywords:** suicide rate, older adult, gender differences, region, Korea

## Abstract

Background: South Korea’s suicide rates are the highest among Organization for Economic Co-operation and Development (OECD) countries, making it one of the most important societal issues in South Korea. Methods: the statistics on causes of death and resident registration central population (RRCP) provided by the National Statistical Office were used to calculate the suicide rate among older adults in Korea. We examined gender (male, female), age (young–old, old–old), and region (urban, rural) by conducting latent growth modeling to estimate changes in the overall older adult suicide rate and verify its relationship. Results: over a period of 8 years, the older adult suicide rate was 104.232 on average in 2010 and it decreased by approximately 10.317 every year, and the rate of decrease gradually slowed down. The initial value of the older adult suicide rate was found to be higher among males, the old–old group, and those living in rural regions. In the quadratic function change rate, only males and the old–old group were statistically significant. Conclusion: in this study, the direction of and the rate of change in the older adult suicide rates and the relationship between gender, age, and region were examined. It is expected that this study will provide basic data to assist in establishing older adult suicide prevention policies, considering the gender, age, and region of the aging population.

## 1. Introduction

South Korea’s suicide rates are the highest among Organization for Economic Co-operation and Development (OECD) countries, making it one of the most important societal issues in South Korea. According to the most recent data from the OECD, the annual suicide rate (the number of deaths per 100,000 persons) was 24.6 in 2019 (OECD, 2022, Suicide rates), which was more than twice the OECD average, 11.2 [1]. Previous studies have indicated that suicide rates are generally accepted to be influenced by the level of the region and to be differed by gender as well as age.

There are gender gaps in terms of suicidal rates between males and females among older adults. The suicide rate of Korean older adults aged 65 and older was 78.9 in males and 24.7 in females per 100,000 in 2017 [2]. This indicates that the suicide rate among males was more than three times higher than that of females (Statistics Korea, 2018). The older adults’ gender difference is higher than the general population; the suicide rate of the general population was 34.9 in males and 13.8 in females [2].

Traditional gender roles as one of various suicide dynamics causes may have had an impact on gender differences in suicide rates among older adults [3,4]. Since Korea has maintained patriarchal norms based on the context of Confucian values, males were more likely to participate in economic activities and females focused on activities for domestic chores and family living [5]. When people’s behavior deviates from the cultural standard and they fail to fulfill the expectations such as gender roles in the Confucian context, increased stress might have an impact on suicide by providing the groundwork for tension between cultural values and reality [6]. Due to the expectation of traditional gender roles, females’ responsibilities are divided into unpaid labor (house chores) and paid employment, lowering the job expectation of females [7]. In this context, studies indicate that socioeconomic status is significantly associated with male suicide rates, but not for female [8,9]. Traditional gender roles have made males exposed to more financial responsibility than females in the household because males are more sensitive to economic difficulties than females even after retirement, which may negatively influence their suicide rates [10].

In addition, there are suicidal characteristics when it comes to age among Korean older adults. They commit suicide at a significantly higher rate than other age groups [11]. South Korea was found to have a specific pattern in suicide (the number of deaths per 100,000 persons in 2019: 27.4 in 15 to 64-year-olds, and 46.6 in those aged 65 years and older) [12]. As well as the pattern between the younger generation and older adults, previous studies indicate the pattern of suicidal ideation (SI) whereby the 65–74 age group had lower SI than the older adults aged 80 and older group did [13]. Furthermore, the pattern of suicidal ideation (SI) showed that the 65–74 age group had lower SI than the older adults in the aged 80 and older group did. The older age group was more likely to have SI than the younger age group [14].

The phenomena above are related to characteristics of geriatric suicide. As older adults get older, they are more likely to experience bio-psycho-social difficulties with aging. Older adults face biological decline such as in their physical activities; psychological problems such as social isolation or loneliness; and social issues such as social isolation after retirement. This multifaceted phenomenon around an individual is reflected in the increment in suicide rates as older adults age [15]. Innamorati et al. (2013) [16] revealed that “old–old” (aged 75 and older) adults who committed suicide are more likely to be bereaved and lived alone than “young–old” (aged between 65 and 74 years old) adults. In addition, they found that “old–old” adults were more likely to experience psychiatric and medical issues than “young–old” older adults. Kim et al. (2018) [17] indicated that illness-related problems among “old–old” older adults were a strong motivator for “old–old” older adult suicide attempters. They found that depression is more prevalent in “old–old” adults than in middle age (45–59 years old) suicide attempters [17].

Changes in suicide rates are affected by regional location and characteristics. Park et al. (2007) [18] argued that suicide rates differ by geographic location. Chan et al. (2015) [19] found that the extent to which a region is urbanized was positively associated with suicide rates among older adults (aged 65 years old and above) from 1998 to 2002, but it was negatively associated with the rates from 2003 to 2007 and 2008 to 2012. Chen et al. (2020) [20] also indicated that the urban–rural suicide ratio is narrowing. They found that the rural–metropolitan suicide rate ratio was 2.1 in 2001, but it was 1.4 in 2016. Cheong et al. (2012) [21] also argued that there are few differences in terms of geriatric suicidal rates between urban and rural areas, but only residential environment and economic status are significantly associated with the suicide rates. However, Lee et al. (2015) [22] claimed that economic status is not greatly different between rural and urban residents in Korea.

Older adults’ increased suicide rates in rural areas may be associated with socioeconomic disadvantage, limited availability or accessibility to medical services, and negative attitudes toward mental illnesses such as depression [20,23,24]. Tarlow et al. (2019) [25] claimed that rural areas’ suicide rates and risks are higher than urban areas among older adults living in the U.S. [25,26]. These explanations may also apply to the Korean context. Rural regions in Korea have been known for the areas with less mobile older age groups, with reduced socioeconomic support and influx from younger generations, as well as declining peer social networks [23,24,27]. The poverty rates, defined by a threshold of half of the median income have increased from 28.7% to 40.8% during the period of 1996 and 2014 [28]. Although the average poverty rate with regards to older adults among OECD member countries is 13%, the recent older Korean adults’ poverty rate had been increasing until 2022 (43.4%) [29]. Although community outreach services effectively reduce the risk of suicide, small rural areas usually experience a lack of medical services, doctors, and psychiatrists. These insufficient support systems are likely to worsen the social isolation of Korean older adults, which results in depression and increased suicide rates in rural areas [24].

Recent changes in suicide rates are also affected by governmental policies. This is consistent with the government’s suicide prevention initiative to decrease the suicide rate since 2010. The government provided pesticide safety boxes to rural areas in 2010 and prohibited the production and distribution of poisonous pesticide, which rapidly decreased the suicide rate in rural areas. Although the sales for the production and distribution of pesticides in the urban areas have a huge impact on the urban suicide rate, the report “World Health Statistics 2017” [30] found that the suicide rate in 2015 (28.4 deaths per 100,000 population) compared to the rate in 2010 (34.1 deaths per 100,000 population) had rapidly decreased because of the government’s prohibition on poisonous pesticide sales. This prevention method reduced the overall suicide rate regardless of gender, age, and location in urban/rural areas.

Previous studies have examined the changes and characteristics of suicide rates differed by influential factors such as age, gender, and region in Korea. However, the studies dealing with suicide rates targeted the younger generation as well as older adults, which only limited to a cross-sectional design [21,27] and did not provide the latest effects of the changes [5,20].

The purpose of this study is to examine changes in geriatric suicide rates that differed by gender, age, and region during the period between 2010 and 2017. Since it is difficult for existing cross-sectional studies to address the changes in suicide rates, this longitudinal study extends existing information about changes in suicidal rates among older adults. In addition, this study may provide valuable insights which address the relationship between suicidal factors and suicidal rates.

## 2. Materials and Methods

### 2.1. Data

The statistics on causes of death and resident registration central population (RRCP) provided by the National Statistical Office were used to calculate the suicide rate among older adults in Korea [2]. We obtained raw data for cause of death statistics from the National Statistical Office’s Micro Data Integrated Service (MDIS) via remote access from 2010 to 2017 [2]. The number of older adult suicides in local governments was calculated using data based on intentional/self-inflicted death as specified by codes (X60-X84) in the 5th Korean standard classification of disease and causes of death. The statistics on causes of death and resident registration central population (RRCP) were provided by the National Statistical Office after they had been coded according to the death notification sent to the administrative agency. Gender, age, and cause of death are listed in the death notification form. RRCP data were used to compute the number of older adult suicides, as well as the older adult suicide rate for each local government. As a result, this study examined the suicide rates of older adults in 252 cities, counties, and districts from 2010 to 2017.

### 2.2. Variables

The dependent variable in this study is the older adult suicide rate. Using the cause of death statistics and RRCP data, the suicide rate was calculated for every 100,000 people in each city, county, and district from 2010 to 2017 using the formula ‘(number of suicides/mid-year population) × 100,000′. The independent variables used in this study were gender, age, and region. For analysis, males were categorized as 0, and females were coded as 1. Young–old (aged 65 to 74 years old) was coded as 0, while old–old (aged 75 and older) was coded as 1. In terms of region, urban areas were classed as 0 and rural areas were classified as 1.

### 2.3. Statistical Analysis

In this study, SPSS 25.0 and M-plus 8.0 programs were used for data handling and research model analysis. First, a descriptive statistical analysis was conducted to identify the characteristics of the older adult suicide rate by gender, age, and region. Second, latent growth modeling was conducted to estimate changes in the overall older adult suicide rate and to verify the relationship between gender, age, and region. Finally, TLI (Tucker–Lewis index), CFI (comparative fit index), and RMSEA (root mean square error of approximation) were used to determine model fit.

## 3. Results

### 3.1. Descriptive Statistics

Based on the analysis of the older adult suicide rate from 2010 to 2017, it was found that overall, the rate declined over time regardless of gender, age, or region (Table 1, Figure 1, Figure 2 and Figure 3). The suicide rate among male older adults reduced from 148.99 rate of deaths in 2010 (Standard Deviation [SD] = 99.90) to 86.34 rate of deaths in 2017 (SD = 63.17), and suicide among females decreased from 55.38 rate of deaths in 2010 (SD = 46.79) to 24.69 rate of deaths in 2017 (SD = 24.14). In the young–old group, the suicide rate dropped from 73.81 in 2010 (SD = 62.94) to 35.80 in 2017 (SD = 34.00), while suicide rates among the older adults dropped from 130.56 in 2010 (SD = 104.81) to 75.22 in 2017 (SD = 67.40). The suicide rates among males and the old–old older adults were found to be greater and fell more rapidly over time than female and young–old suicide rates (Figure 1 and Figure 2). In the case of the older adults in rural regions, the suicide rate was found to be essentially identical to that of the older adults in urban areas, although it declined more in 2017 than in 2010 (Figure 3).

The model was examined in two stages in this study. The initial value and change rate were estimated in the first stage using unconditional model analysis. The association between the change in the older adult suicide rate and gender, age, and region was examined in the second stage using conditional model analysis, based on the initial value and the change rate obtained in the first step.

#### 3.1.1. Analysis of Unconditional Model

To explain the changes in the geriatric suicide rate, an unconditional model analysis was performed before proceeding with the conditional model analysis. Furthermore, the no growth model, linear growth model, and quadratic growth model were all examined in order to find the best change pattern using the unconditional model. The fitness of the quadratic growth model was χ2 = 123.947 (*p* < 0.001), CFI = 0.971, TLI = 0.970, RMSEA = 0.060 for the older adult suicide rate. After statistical analysis, it was revealed that the quadratic growth model described the change in the suicide rate of older adults better than the no growth or the linear growth model; therefore, the quadratic growth model was adopted for the final analysis (Table 2).

The average initial value of the suicide rate among older adults in 2010 was 104.232 (*p* < 0.001), the linear rate of change was 10.317 (*p* < 0.001), and the quadratic function rate of change was 0.516 (*p* < 0.001); all of which were statistically significant (Table 3). In other words, as shown in Figure 4, the older adult suicide rate declines over time, but at a slower rate. Moreover, the variance was statistically significant with the initial value of 4480.481 (*p* < 0.001), the linear change rate of 166.694 (*p* < 0.01), and the quadratic function change rate of 1.893 (*p* < 0.05). This demonstrates that across local governments, the suicide rate of older adults differs considerably at both the initial level and at the rate of change.

#### 3.1.2. Analysis of Conditional Model

Table 4 shows the result of the conditional model analysis by examining the coefficient of the path between variables. In the conditional model analysis, we examined how gender, age, and region affect the initial value and the change rate of the older adult suicide rate. The result of the model fit analysis was χ2 = 137.579 (*p* < 0.001), CFI = 0.979, TLI = 0.974, RMSEA = 0.048. In the analysis, gender (B = −96.314, *p* < 0.001), age (B = 57.946, *p* < 0.001), and region (B = 10.932, *p* < 0.001) were found to have a statistically significant effect on the initial value of the older adult suicide rate. In other words, the initial value of the older adult suicide rate was found to be higher among males, the old–old group, and those living in rural regions. Just as with the initial value of the older adult suicide rate, the rate of the linear growth model appeared to have statistical significance; gender (B = 8.172, *p* < 0.001), age (B = −6.002, *p* < 0.01), and region (B = −4.811, *p* < 0.05).

However, in the quadratic function change rate, only gender (B = −0.524, *p* < 0.05) and age (B = 0.501, *p* < 0.05) appeared to have a statistically significant effect. In other words, it was found that the rate of decline among the older adults’ suicide rate increased in the males and the old–old group. This suggests that the suicide rate decreases more rapidly in the male and the old–old group than in the female and young–old group. The rate of change of the quadratic function of the older adult suicide rate was not affected by region.

## 4. Discussion

The purpose of this study is to examine the association between the change in the suicide rate among older adults by gender, age, and region during a period of 8 years longitudinally from 2010 to 2017. The main findings of this study are summarized and discussed as follows. First, the quadratic growth model was shown to be the most appropriate model for the patterns in the older adult suicide rate from 2010 to 2017. The older adult suicide rate was 104.232 on average in 2010, which corresponds to the initiation of the study, and it has decreased by approximately 10.317 every year since. Over a period of 8 years, the older adult suicide rate decreased over time, and the rate of decrease gradually slowed down, showing similar results as the previous studies [31].

However, the result of the initial value and the rate of change among older adult suicide rates differed significantly by district. Indeed, according to Jeong (2019) [32], who examined the variations in the older adult suicide rate by city, county, and district from 2010 to 2016 in Korea, certain regions had a number of older adult suicides equaling 30, while others had more than 100. In addition, the previous study indicates a complexity in the change in suicide rate among older adults in a residential area in Korea; there were three types of variations in the suicide rate of older adults (medium-level stable type, high-level stable type, and declining type) [32]. These findings show that a community-centered approach strategy is critical for preventing older adult suicide [33], and that policy intervention should be segmented based on the type of change in the older adult suicide rate by the local government. According to the Relevant Ministry Joint in Korea (2019) [34], this approach strategy was announced as a project to be pursued in order to provide personalized policy formulations at regional and district levels, based on an analysis of policy success and failure factors for regions with varied suicide rates. Although the “Five-Year National Suicide Death Analysis Results Report” published by the Ministry of Health and Welfare (2021) [35] included regions with particularly high or particularly low rates of suicide in old age, there was inadequate discussion on causes or interventions of suicide rates. Therefore, it is necessary to validate regional characteristics through follow-up studies that identify regions with a consistently high older adult suicide rate by supporting the local government’s suicide prevention policies based on evidence from this study.

Secondly, gender was discovered to be a significant influencing factor on both the initial value and the change rate of the older adult suicide rate as a result of the conditional model analysis. This suggests that in the aging population, male suicide rates are higher than female suicide rates, and the declining trend slows down as the population ages. As gender is one of the most important predictors of suicide in the aging population [36], understanding the role of gender-specific responses may assist in mitigating the risk of suicide among older adults [37]. To begin with, it is well known that low educational attainments and low economic status are associated with suicide ideation among older males [9]. In part, in the traditional Korean context, this is because men tend to have greater economic responsibility and burden for their families, which may have led to them having an increased risk of suicide [9]. In their later years, although males receive income security support such as the National Pension Service, income variations decrease to roughly half of the middle-aged income, causing economic instability and this also became another risk factor for an increase in suicidal thoughts [38]. In this regard, it is imperative that older men have access to the senior employment program within their local communities so that they may participate in economic activities even after retirement without feeling socially isolated. It has been found that participation in the senior employment program enhanced self-efficacy and life satisfaction among older individuals [38,39]. Furthermore, once the effectiveness of the New Welfare Policy Model, Seoul Special City (2021) [40], a new welfare policy model promoted by the Seoul Metropolitan Government in 2022, is validated, a government review on policy alternatives related to income security that will alleviate poverty problems among the aging population in the future is required.

Despite the fact that the suicide rate among older females was initially lower than that of males, the downward trend was found to be mild. According to Joo and Lee (2014), suicide among older adults shows a tendency to have a phenomenon that spreads both temporally and spatially, and older females showed more vulnerability to this diffusion effect than older males. In other words, although the suicide rate of the older females is significantly lower than that of the older males, they show a higher tendency to be affected by the spread of suicides in the surrounding area, suggesting that a differentiated strategy should be established in implementing suicide prevention policies for older females. As a result, it is expected that the Ministry of Health and Welfare’s “National Suicide Death Analysis Report” would also provide fundamental statistics, such as specific information on suicide-related information and the leading causes of suicide by gender. Moreover, the local administrations should also demand more specific involvement in policy-making in autonomous districts, considering the characteristics of the aging population (gender ratio, etc.).

Third, the older adult suicide rate was initially greater in the old–old group, and the pace of decline was relatively rapid as time passed. Physical and functional characteristics such as dental issues, pain, and subjective health condition are indicated as substantial risk factors for suicidal ideation among old–old older adults as opposed to young–old older adults [11]. Physical health has a high association with the individual’s ability to live independently; deterioration in health and complaints of pain reduce the older adults’ life satisfaction in later life, contributing to a rise in the suicide rate [41]. A study comparing the characteristics of suicide attempters by age discovered that the latter-stage older adults were not only physically more susceptible than the former-stage older adults but they also had stronger suicidal ideation and a lower likelihood of asking for help from others [17]. Therefore, it is suggested that the local governments with a high prevalence of geriatric suicides should first provide home visit health care services, identify the types of services they need, and customize physical, mental, and social health care programs for individuals. In the city of Seoul, for instance, visiting nurses and social workers have been providing integrated community health management services to older adults since 2015 for the purposes of health management, disease education, and psychological support. In the program, 53.3% of older adult participants received depression and suicide impact assessments, and 33.1% of them were referred to medical facilities, which appears to contribute to disease prevention and health promotion of the older adults centered on the local community rather than institutions or hospitals [42].

On the other hand, marital conflict has been reported to influence suicidal ideation among the young–old group [43], implying that as the amount of time spent together after retirement increases, so does the problem of marital conflict. However, this information alone cannot explain the factors that influence suicide rates among older adults. It is, therefore, necessary for the aforementioned visiting health management program managers (such as nurses and social workers) to collect and analyze data about common aspects found among the young–old aging population while performing their respective duties.

Fourthly, the finding that older adults in rural regions have a higher suicide rate is consistent with the previous studies that showed the suicide rate is greater in rural areas than in metropolitan cities [32,44]. In rural regions, it is reported that the residents are resistant to asking for help or receiving treatment due to the conservative and closed nature of the region [45]. In addition, the ratio of the social welfare budget to the population of older adults is smaller in rural areas where the suicide rate is higher than in major cities [44]; previous research has observed a relationship between the ratio of the social welfare budget to the population of older adults with the suicide rates [32,46]. As a result, in a rural region, it is necessary to benchmark domestic cases [47], such as taxi voucher support projects that support and increase the accessibility of the welfare facilities, and afterward improve mobility to a level similar to older adults living in the urban areas. Therefore, it is suggested that the local governments should prioritize social welfare policy expenditures and efforts to protect the aging population who live in rural areas with a high suicide rate.

Finally, in this study, the rate of change in the geriatric suicide rate showed that regional variables were not significant in the rate of change in the quadratic function. The regional differences and complexity that are not statistically revealed in the analysis, unlike the individual level, are inherent, so that the regional effect of reducing older adult suicide rates can disappear or be offset. Even in rural areas where the suicide rate among older adults is particularly low, there was a difference in the suicide rate based on the aspects of the level of neighborhood interactions [45]. This can be interpreted as it being hard to simply divide into urban and rural areas to promote or alleviate the suicide rate of the aging population as the data is far more complex due to the existence of regional variations and complexity within the regional characteristics. It is, therefore, necessary to discuss in-depth the relationship between change in the suicide rate among older adults and the region through follow-up research.

The limitations and significance of this study are as follows. This study is significant as a quantitative study based on secondary data since it examined the initial value and rate of change in the older adult suicide rate by gender, age, and region to estimate the overall change over time in geriatric suicide rates. In spite of this, the study emphasizes the need to establish a policy specifically tailored to older adult suicide prevention considering regional characteristics, however, it could not derive concretely what factors have contributed to the reduction in suicide rates among older adults other than the previously known prohibition on the sale of pesticides. Moreover, since the local characteristics in this study are based on an analysis of two categories (urban and rural), the conclusions need to be interpreted with caution, because the suicide rate of older adults may differ not only depending on the region but also depending on the individual’s gender and age, and the physical and social environment to which the individual belongs. However, it has been established in earlier studies that there are complications and disparities in geographical characteristics and suicide rates among the aging population. Therefore, further research is needed to fully comprehend the underlying significance behind the decline in the suicide rates among older adults. It is anticipated that by examining the effects of regional influences with a focus on Jeollanam-do, which has a low geriatric suicide rate, and areas where the older adult suicide rates are low or high, the insights as to the variables that promote or alleviate suicide by region can be discovered.

## Figures and Tables

**Figure 1 healthcare-10-02333-f001:**
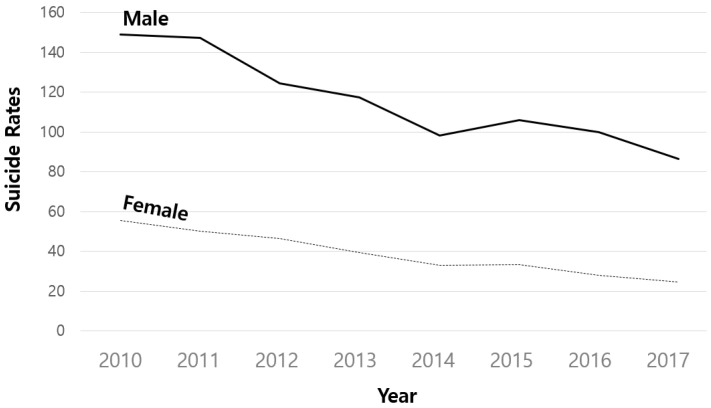
Suicide rate of older adults by gender.

**Figure 2 healthcare-10-02333-f002:**
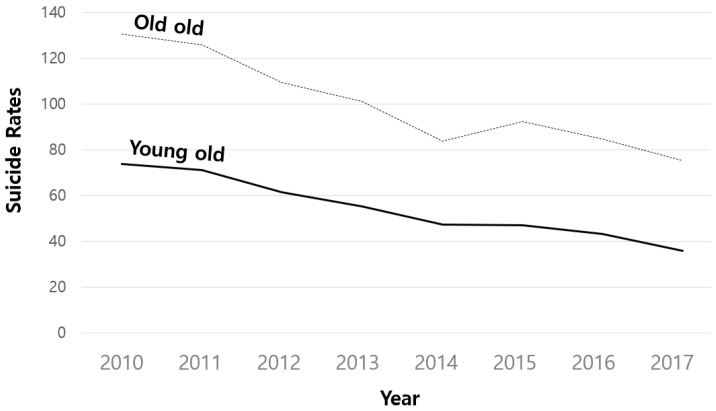
Suicide rate of older adults by age.

**Figure 3 healthcare-10-02333-f003:**
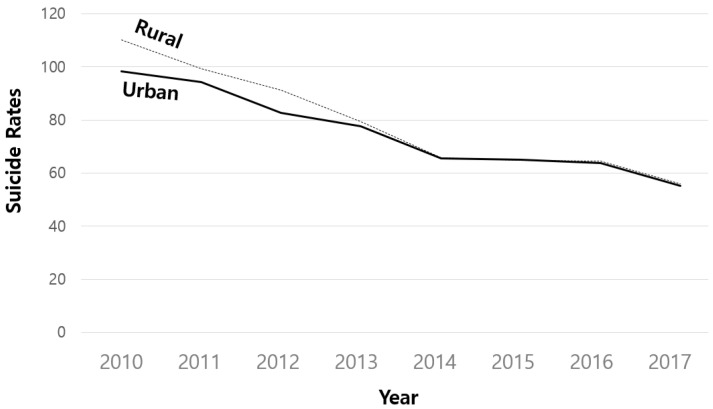
Suicide rate of older adults by region.

**Figure 4 healthcare-10-02333-f004:**
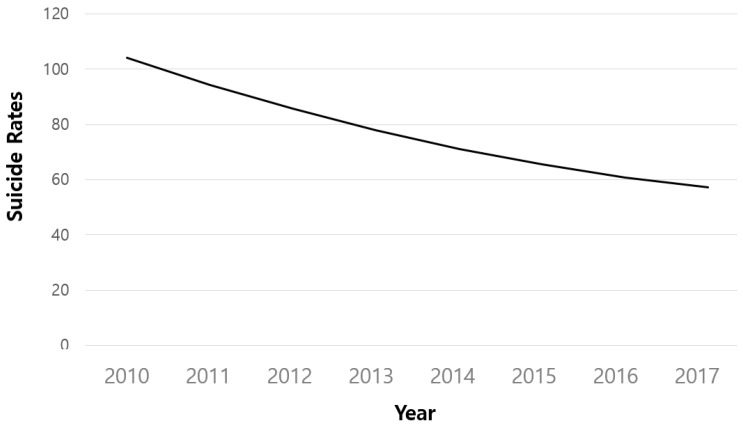
Quadratic function estimate of the older adult suicide rate 3.2. Analysis of the study model.

**Table 1 healthcare-10-02333-t001:** Suicide rate of older adult by gender, age, and region (2010–2017).

Year	Gender				Age				Region			
	Male (*n* = 252)		Female (*n* = 252)		Young–Old (*n* = 252)		Old–Old (*n* = 252)		Urban (*n* = 170)		Rural (*n* = 82)	
	M	SD	M	SD	M	SD	M	SD	M	SD	M	SD
2010	148.99	99.90	55.38	46.79	73.81	62.94	130.56	104.81	98.32	78.99	110.17	111.35
2011	147.13	93.63	50.21	42.92	71.31	57.73	126.03	102.37	94.38	81.11	99.26	99.40
2012	124.49	84.33	46.45	38.86	61.49	50.70	109.45	89.16	82.68	67.17	91.23	92.30
2013	117.24	75.92	39.37	34.97	55.36	48.22	101.25	81.51	77.78	66.10	79.38	79.61
2014	98.11	65.15	33.12	29.55	47.49	41.63	83.74	69.58	65.48	56.49	65.89	67.09
2015	106.08	75.29	33.17	28.56	46.96	41.06	92.28	80.20	64.95	63.91	64.79	74.52
2016	99.94	68.60	28.08	24.05	43.37	38.50	84.65	74.40	63.71	57.61	64.64	72.22
2017	86.34	63.17	24.69	24.14	35.80	34.00	75.22	67.40	55.26	51.39	56.04	66.93

Note: M (mean), SD (standard deviation).

**Table 2 healthcare-10-02333-t002:** Model Fit of the Unconditional Model.

Model	χ2	df	CFI	TLI	RMSEA
No Growth Model	824.012 ***	34	0.763	0.805	0.152
Linear Growth Model	161.136 ***	31	0.961	0.965	0.065
Quadratic Growth Model	123.947 ***	27	0.971	0.970	0.060

*** *p* < 0.001.

**Table 3 healthcare-10-02333-t003:** Mean and Variance of initial score and rate of change of Unconditional Model.

Variables	Mean		Variance	
	Estimate	S.E.	Estimate	S.E.
Initial value	104.232 ***	2.667	4480.481 ***	335.365
Linear rate of change	−10.317 ***	1.034	166.694 **	55.106
Quadratic rate of change	0.516 ***	0.128	1.893 *	0.855

* *p* < 0.05, ** *p* < 0.01, *** *p* < 0.001.

**Table 4 healthcare-10-02333-t004:** Path Coefficient of Study Model.

Path between Variables			Coef.	S.E.
Gender (ref. Male)	→	Initial value	−96.314 ***	3.972
Age (ref. Young old)	→	Initial value	57.946 ***	3.967
Area (ref. Urban)	→	Initial value	10.932 *	4.231
Gender (ref. Male)	→	Linear rate of change	8.172 ***	2.032
Age (ref. Young old)	→	Linear rate of change	−6.002 **	2.029
Area (ref. Urban)	→	Linear rate of change	−4.811 *	2.162
Gender (ref. Male)	→	Quadratic rate of change	−0.524 *	0.254
Age (ref. Young old)	→	Quadratic rate of change	0.501 *	0.253
Area (ref. Urban)	→	Quadratic rate of change	0.474	0.270

* *p* < 0.05, ** *p* < 0.01, *** *p* < 0.001; → indicates route.

## Data Availability

The datasets generated during and/or analyzed in this study are publicly available upon request from: https://kostat.go.kr/portal/eng/surveyOutline/5/1/index.static (accessed on 1 April 2022).

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
