# Peer review of "Changes in the Suicide Rate of Older Adults According to Gender, Age, and Region in South Korea from 2010 to 2017"

_healthcare, 2022, doi:10.3390/healthcare10112333_

Round 1

Reviewer 1 Report

Thank you for the opportunity to review this paper. It provides interesting and important information on the decrease of suicide rates of the older population in South Korea.

My suggestions are mainly around the positioning of the findings into suicide prevention literature.

Abstract

Conclusion a bit a vague.

Background
Line 44 - don't quite understand the sentence
Line 68 - don't quite understand the sentence
Line 72 - don't quite understand the sentence
Line 99 - please review sentence

A paragraph on suicide prevention initiatives in South Korea would position where this study sits. In 2010, did new suicide prevention initiatives contribute to the decrease over the coming years? If not, that is also important to mention.

Maybe less paragraphs about gender.

Method
Appears solid.

Discussion
Line 234 - don't quite understand sentence

Line 263 - There is incongruence between males going back to work and their health declining. Maybe linkage between these concepts would be useful. Maybe older people want a tool box, e. g. opportunity to go back to work, social opportunites, link with community, etc.

Line 274 - 276 - Don't quite understand sentence

I was wondering if qualitative studies that include young-old and old-old people could capture some of their views on resilience and strategies that they find helpful. I liked the suggestion about home visit health care services, it might need review/evaluation on effectiveness and suicides?

Line 298 - appears a weak argument about marital relationships and second half of paragraph does not link well with earlier focus on marital status.

It is unclear what the authors stipulate as the reason for the decreasing suicide rate. This information can link to paragraph in the background section as to what suicide prevention strategies have been done. If this information is covered in prior research, please refer to it.

Line 306 - how do you know your grouping of young-old vs old-old is optimal (especially with increasing age)? Could there be other groupings?

How are suicides determined, what is the quality of the data, how is data collected and coded - does it vary across districts? Has that changed over time? Is that a limitation?

Is there a place to acknowledge that life gets harder as we age - and should older people have a say in their death, e.g. euthanasia? Or, that researchers who study older people might need to involve older people into their research team so findings/outcomes actually make a difference to this group of people and is not guided by people who think they know what older people want?

Adding more context to the article would in my opinion make it of high interest to readers.

Author Response

Abstract

Conclusion a bit a vague.

--> Thank you for your suggestion. We modified conclusion under the Abstract for better understanding.

Background
Line 44 - don't quite understand the sentence  

--> Thank you for your suggestion. We modified the sentence for better understanding.

Line 68 - don't quite understand the sentence

--> Thank you for your suggestion. We modified the sentence for better understanding.

Line 72 - don't quite understand the sentence

-->Thank you for your suggestion. We modified the sentence for better understanding.

Line 99 - please review sentence  

-->Thank you for your suggestion. We modified the sentence for better understanding.

A paragraph on suicide prevention initiatives in South Korea would position where this study sits. In 2010, did new suicide prevention initiatives contribute to the decrease over the coming years? If not, that is also important to mention. ->

--> Thank you for the feedback. Yes, the new suicide prevention initiatives contribute to the decrease over the coming years. We added the paragraph between lines 122 and 132 as follows.

Recent changes in suicide rates are also affected by governmental policies. This is consistent with the government’s suicide prevention initiative to decrease the suicide rate since 2010. Government provided pesticide safety box to the rural areas in 2010 and prohibited production and distribution of poisonous pesticide, which rapidly dropped the suicide rate in the rural areas. Although the sales for the production and distribution of the pesticide in the urban areas has a huge impact on the urban suicidal rate, the report “World Health Statistics 2017” (World Health Organization, 2017) found that the suicide rate in 2015 (28.4 deaths per 100,000 population) compared to the rate in 2010 (34.1 deaths per 100,000 population) was rapidly decreased because of the government’s prohibition against poisonous pesticide sale. This prevention reduced the overall suicide rate regardless of gender, age, and urban/rural areas.

Maybe less paragraphs about gender.

--> Thank you for your suggestion. We have shortened the part on gender in the introduction part.

Method
Appears solid.

--> Thank you.

Discussion
Line 234 - don't quite understand sentence  

--> Thank you for pointing that out. We made the sentence clearer to assist in better understanding readers. 

Line 263 - There is incongruence between males going back to work and their health declining. Maybe linkage between these concepts would be useful. Maybe older people want a tool box, e. g. opportunity to go back to work, social opportunites, link with community, etc.  

--> We are glad that you have pointed it out. We have removed ambiguity and added senior employment programs as suggested. 

Line 274 - 276 - Don't quite understand sentence

--> We appreciate your point. We have modified the sentence for better understanding as below:

According to Joo and Lee (2014), suicide among older adults shows a tendency to have a phenomenon that spreads both temporally and spatially, and older females showed more vulnerability to this diffusion effect than older males. In other words, although the suicide rate of the older female is significantly lower than that of the older male, they show a higher tendency to be affected by the spread of suicides in the surrounding area, suggesting that a differentiated strategy should be established in implementing suicide prevention policies for older females.

I was wondering if qualitative studies that include young-old and old-old people could capture some of their views on resilience and strategies that they find helpful. I liked the suggestion about home visit health care services, it might need review/evaluation on effectiveness and suicides?

--> Thank you for your suggestion. We have added the explanation of the home visit health care and its effectiveness.

Line 298 - appears a weak argument about marital relationships and second half of paragraph does not link well with earlier focus on marital status. 

--> Thank you for your suggestion. We have modified the sentence indicating the limitation of explaining the factors that influence suicide rates among older adults.

It is unclear what the authors stipulate as the reason for the decreasing suicide rate. This information can link to paragraph in the background section as to what suicide prevention strategies have been done. If this information is covered in prior research, please refer to it.

--> We appreciate your suggestion. We added some sentences in limitation as follows:

In spite of this, the study emphasizes the need to establish a policy specifically tailored to older adult suicide prevention considering regional characteristics, however, it could not derive concretely what has contributed to the reduction in suicide rates among older adults other than the previously known prohibition of the sales of pesticides.

Line 306 - how do you know your grouping of young-old vs old-old is optimal (especially with increasing age)? Could there be other groupings? -> à We appreciate your feedback. Based on your opinion, we have indicated that there is a limitation. Therefore, the suggestion that the aging group should be segmented by age (young-old vs. old-old) was removed.

How are suicides determined, what is the quality of the data, how is data collected and coded - does it vary across districts? Has that changed over time? Is that a limitation?

--> We are glad that you pointed this out. Suicide data can be extracted from statistics on the cause of death, and we added specific details under the ‘2. Materials & Methods-2.1. Data' section.

Is there a place to acknowledge that life gets harder as we age - and should older people have a say in their death, e.g. euthanasia? Or, that researchers who study older people might need to involve older people into their research team so findings/outcomes actually make a difference to this group of people and is not guided by people who think they know what older people want?

--> Thank you for your clarification. For old-old group, we have indicated that the physical health has a high association with the individual’s ability to live independently; the deterioration of health and complaints of pain reduces the older adults’ life satisfaction in later life, contributing to a rise in the suicide rate (line 303-306). However, the suggestion that the aging group should be segmented by age (young-old vs. old-old) was removed.

Reviewer 2 Report

Really interesting article. The sections are respected and the statistical analysis was correctly performed.

I appreciated the inclusion of the graphs in the results section.

However, the bibliography needs to be expanded with additional articles pertaining to various suicide dynamics. I propose these two articles:

- A special case of suicide enacted through the ancient Japanese ritual of Jigai. The American journal of forensic medicine and pathology, 35(1), 8-10. https://doi.org/10.1097/PAF.0000000000000070

-Determined to die! Ability to act upon multiple self-inflicted gunshot wounds to the head. The Cook County Medical Examiner's Office experience (2005-2012) and review of the literature. Journal of forensic sciences, 60(5), 1373-1379. https://doi.org/10.1111/1556-4029.12780

Author Response

Really interesting article. The sections are respected and the statistical analysis was correctly performed.

I appreciated the inclusion of the graphs in the results section.

However, the bibliography needs to be expanded with additional articles pertaining to various suicide dynamics. I propose these two articles:

- A special case of suicide enacted through the ancient Japanese ritual of Jigai. The American journal of forensic medicine and pathology, 35(1), 8-10. https://doi.org/10.1097/PAF.0000000000000070 

--> We appreciate your suggestion. We have added the recommended reference in the introduction as below:

Traditional gender roles as one of various suicide dynamics causes may have had an impact on gender differences in suicide rates among older adults [3,4]

-Determined to die! Ability to act upon multiple self-inflicted gunshot wounds to the head. The Cook County Medical Examiner's Office experience (2005-2012) and review of the literature. Journal of forensic sciences, 60(5), 1373-1379. https://doi.org/10.1111/1556-4029.12780

-->  We appreciate your suggestion. We have added the recommended reference in the introduction as below:

Traditional gender roles as one of various suicide dynamics causes may have had an impact on gender differences in suicide rates among older adults [3,4]

Reviewer 3 Report

Authors decided to investigate the association between the change in the  suicide rate among older adults by gender, age, and region during a period of 8 years. It was longitudinal survey covered the period from 2010 to 2017.   A valuable approach by the authors is to relate their own research results to those obtained by other authors in the past.  They added own results and  made additional remarks.   Four extended conclusions were formulated on the basis of the results of investigation, with recommendations for local governments what to do to diminish the suicide rates.  Groups of older people requiring special attention by local governments were males, old-old (aged 75 and older) and those living in rural regions. The authors emphasised (and they are absolutely right) the necessity  of segmentation  older adults into age groups (young-old vs. old-old) and  the need to conduct researches separately for both groups.

Two remarks:

·         To explain the change in the birth rate,(…)’ (line 180) – it is not clear why the Authors start the point 3.2.1. Analysis of Unconditional Model with the birth rate?

·         Characteristics of suicidal characteristics of Korean older people and  the roles of traditional gender roles supported by Confucianism (Confucian values) (line 54-55) is insufficient for a Christian reader.  In Christianity, suicide is a grave sin and this might an impact on the  suicide rate of older people.  Give us one-two sentences how Confucianism  treats suicide as a  defensible or acceptable act.

Author Response

Two remarks:

To explain the change in the birth rate,(…)’ (line 180) – it is not clear why the Authors start the point 3.2.1. Analysis of Unconditional Model with the birth rate?

--> We appreciate your clarification. We have modified the birth rate to the suicide rate.

Characteristics of suicidal characteristics of Korean older people and  the roles of traditional gender roles supported by Confucianism (Confucian values) (line 54-55) is insufficient for a Christian reader.  In Christianity, suicide is a grave sin and this might an impact on the  suicide rate of older people.  Give us one-two sentences how Confucianism  treats suicide as a  defensible or acceptable act.

--> We appreciate your suggestion. We have added a sentence for an abundant explanation as below:

 When people’s behavior deviates from the cultural standard and they fail to fulfill the expectations such as gender roles in the Confucian context, increased stress might have an impact on suicide by providing the groundwork for tension between cultural values and reality [6]
